# GABA_A presynaptic inhibition regulates the gain and kinetics of retinal output neurons

**Jenna Nagy[1,2,3], Briana Ebbinghaus[2,4,5], Mrinalini Hoon[1,2,4], Raunak Sinha[1,2,4]\***

[1]Department of Neuroscience, University of Wisconsin, Madison, United States; [2]McPherson Eye Research Institute, University of Wisconsin, Madison, United States; [3]Cellular and Molecular Pathology Training Program, University of Wisconsin, Madison, United States; [4]Department of Ophthalmology and Visual Sciences, University of Wisconsin, Madison, United States; [5]Neuroscience Training Program, University of Wisconsin, Madison, United States

**Abstract** Output signals of neural circuits, including the retina, are shaped by a combination of excitatory and inhibitory signals. Inhibitory signals can act presynaptically on axon terminals to control neurotransmitter release and regulate circuit function. However, it has been difficult to study the role of presynaptic inhibition in most neural circuits due to lack of cell type-specific and receptor type-specific perturbations. In this study, we used a transgenic approach to selectively eliminate GABA_A inhibitory receptors from select types of second-order neurons – bipolar cells – in mouse retina and examined how this affects the light response properties of the well-characterized ON alpha ganglion cell retinal circuit. Selective loss of GABA_A receptor-mediated presynaptic inhibition causes an enhanced sensitivity and slower kinetics of light-evoked responses from ON alpha ganglion cells thus highlighting the role of presynaptic inhibition in gain control and temporal filtering of sensory signals in a key neural circuit in the mammalian retina.

**\*For correspondence:**
raunak.sinha@wisc.edu

**Competing interests:** The authors declare that no competing interests exist.

## Introduction

A common motif by which inhibition acts in neural circuits is at the axon terminals of presynaptic neurons where it regulates synaptic release and controls the input-output relationship of a neural circuit (*Fink et al., 2014*; *MacDermott et al., 1999*). This motif of inhibition called presynaptic inhibition is widely used in the retina and is mediated by inhibitory retinal interneurons called amacrine cells (ACs) (*Diamond and Lukasiewicz, 2012*; *Eggers and Lukasiewicz, 2011*; *Eggers et al., 2007*; *Jadzinsky and Baccus, 2013*). ACs make synaptic contacts on the axon terminals of glutamatergic second-order neurons called bipolar cells (BCs) which relay rod and cone photoreceptor signals to retinal output neurons called retinal ganglion cells (RGCs) (*Demb and Singer, 2015*). Axon terminals of 'ON' BCs – that depolarize in response to a light increment – and 'OFF' BCs – that hyperpolarize in response to a light increment – each stratify at different retinal laminae (*Demb and Singer, 2015*; *Hoon et al., 2014*). Of note, different BC types are also used to route dim light (rod bipolar; ON type) and day light (cone bipolar; ON and OFF types) visual signals (*Euler et al., 2014*). In this study, we explored the role of retinal presynaptic inhibition in regulating the output of one of the most well-characterized retinal circuits that use the ON sustained alpha GC (ONα GC) (*Grimes et al., 2014b*; *Murphy and Rieke, 2006*; *Schwartz et al., 2012*; *van Wyk et al., 2009*). ONα GCs depolarize and respond with an increase in action potential firing to light increments (*Grimes et al., 2014b*; *Murphy and Rieke, 2006*; *van Wyk et al., 2009*). This ONα GC pathway is not only the most sensitive dim light retinal pathway (*Smeds et al., 2019*): rod photoreceptors -> rod bipolar cells (RBCs) -> AII amacrine -> ON cone bipolar cell (CBC) -> ONα GC; but also one that routes visual signals

directly from cone photoreceptors via ON CBCs for mediating day light vision (**Demb and Singer, 2015**; **Grimes et al., 2014b**; **Schmidt et al., 2014**; **Figure 1A**).

Previous studies have extensively characterized the molecular composition and expression of GABA/Glycine receptors mediating presynaptic inhibition at axon terminals of ON and OFF BCs (**Eggers and Lukasiewicz, 2006**; **Fletcher et al., 1998**; **Hoon et al., 2014**; **Hoon et al., 2015**; **Lukasiewicz et al., 2004**; **Matthews et al., 1994**; **Wässle et al., 1998**). RBCs and ON CBCs predominantly express $GABA_A$ and $GABA_C$ receptors at their axon terminals, whereas OFF CBCs predominantly express $GABA_A$ and glycine receptors (GlyR) at their axon terminals (**Eggers and Lukasiewicz, 2006**; **Fletcher et al., 1998**; **Hoon et al., 2015**; **Lukasiewicz et al., 2004**; **Wässle et al., 1998**). In the primary rod (dim-light) pathway, presynaptic inhibition has been particularly well characterized where a specialized AC type, the A17 AC, makes GABAergic feedback synapses on RBC axon terminals (**Grimes et al., 2014a**; **Grimes et al., 2010**; **Grimes et al., 2015**). This A17-mediated feedback inhibition has been proposed to improve the signal-to-noise ratio of the feedforward excitatory output from RBC to AII amacrine cell near visual threshold, extend the luminance range over which RBC-AII synapses compute contrast gain, and mediate center-surround inhibition (**Grimes et al., 2015**; **Oesch and Diamond, 2019**; **Völgyi et al., 2002**). In addition, a few studies have shown that perturbing $GABA_C$ receptor-mediated presynaptic inhibition alters the dynamic range of the light-evoked responses of RGCs for both rod and cone-mediated signaling (**Oesch and Diamond, 2019**; **Pan et al., 2016**; **Sagdullaev et al., 2006**).

Our understanding of how presynaptic inhibition shapes signaling at the BC output synapse largely comes from pharmacological manipulations or transgenic mutant mice globally lacking inhibitory receptor types (**Eggers and Lukasiewicz, 2006**; **Eggers and Lukasiewicz, 2011**; **Oesch and Diamond, 2019**; **Pan et al., 2016**; **Sagdullaev et al., 2006**). These approaches affect the entire retinal circuitry and thus lack the resolution required to perturb presynaptic inhibition in a cell type-specific and circuit-specific manner. In addition, such methods make it difficult to parse out the role of $GABA_A$ receptor-mediated presynaptic inhibition because, besides BCs, $GABA_A$ receptors are also expressed on ACs that can participate in serial inhibitory circuits between ACs that in turn contact BC terminals (**Eggers and Lukasiewicz, 2011**; **Wässle et al., 1998**). In fact, immunolabeling against the dominant subunit of $GABA_A$ receptors shows a dense expression, not specific to a single cell class but localized throughout the retinal synaptic layer that makes it particularly difficult to determine the specific contribution of $GABA_A$ receptors in shaping the synaptic output of individual retinal cell types (**Hoon et al., 2015**; **Wässle et al., 1998**; **Figure 1B**). Due to these limitations, most studies investigating the role of GABAergic presynaptic inhibition in shaping the light sensitivity of ON BC synapses have largely been restricted to evaluating $GABA_C$ receptor-mediated inhibition that is specifically localized to BCs (**Oesch and Diamond, 2019**; **Pan et al., 2016**; **Sagdullaev et al., 2006**). This motivated us to use genetic manipulations that selectively eliminate $GABA_A$ receptors from ON BC axon terminals to study the role of $GABA_A$ receptor-mediated presynaptic inhibition in regulating light evoked function in the tractable retinal circuit of the ONα GC with known cell types and a well-established pathway for dim light and day light signals.

## Results

### Selective removal of $GABA_A$ receptors from the axon terminals of rod and ON CBCs

In this study, we focused on $GABA_A$ receptor-mediated inhibition at axon terminals of ON BCs. $GABA_A$ receptors expressed in the axon terminals of ON BCs contain α1 and γ2 subunits (**Hoon et al., 2015**). To specifically eliminate $GABA_A$ receptors from ON BCs we used a *Gabrg2* ($GABA_A$ receptor, subunit gamma 2) floxed mutant mice (**Schweizer et al., 2003**) crossed to an ON BC specific Cre line (*Grm6*-Cre) and a fluorescent reporter line (Ai9/tdTomato) (**Hoon et al., 2015**). This triple transgenic mouse line – Ai9/*Grm6*Cre/*Gabrg2* cKO (henceforth referred to as KO) – has previously been used to study the subunit composition of $GABA_A$ receptors in axon and dendrites of ON CBCs after *Gabrg2* deletion (**Hoon et al., 2015**). Here we determined the $GABA_A$ and $GABA_C$ receptor expression across axon terminals of RBCs which belong to the ON BC class after *Gabrg2* deletion (**Figure 1B–D**). Immunolabeling with $GABA_A$α1 receptor subunit specific antibody clearly showed the selective reduction in $GABA_A$α1 expression from the ON sub-lamina of the inner

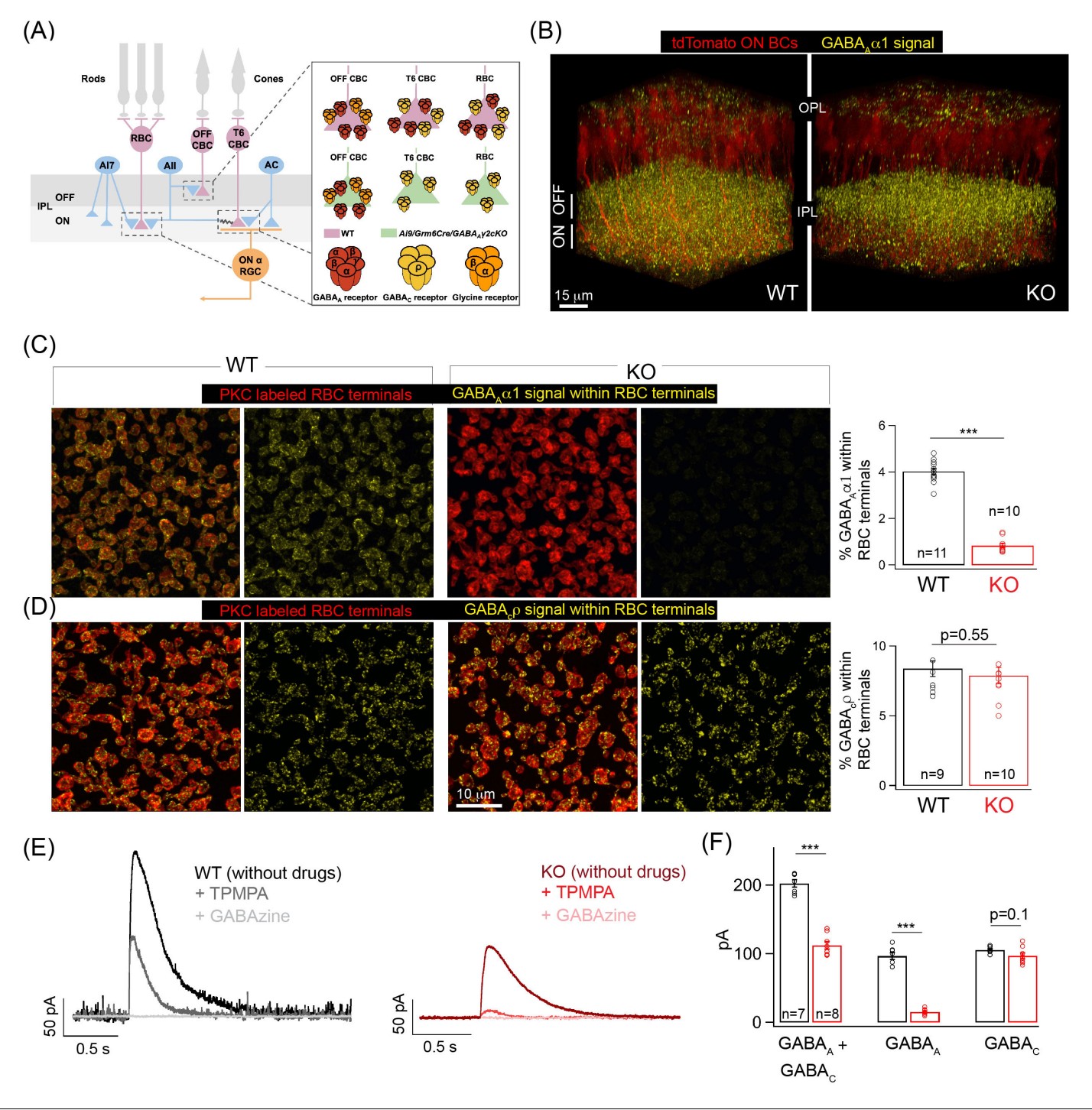

**Figure 1.** Specific elimination of GABA_A receptors from ON bipolar cell (BC) terminals in the Ai9/*Grm6*Cre/*Gabrg2* conditional knockout (KO) mouse. (A) Schematic illustrating the receptor composition of presynaptic inhibition across rod (RBC) and cone bipolar cell (CBC) axon terminals in wildtype adult littermate control (WT) and KO retina; T6 CBC refers to Type 6 ON CBC (B) α1-subunit-containing GABA_A receptor (GABA_A α1) immunolabeling (yellow) and ON BC labeling (tdTomato; red) in outer and inner plexiform layers (OPL and IPL respectively) of WT retina and Ai9/*Grm6*Cre/*Gabrg2* (KO) retina. In the KO, GABA_Aα1 immunofluorescence is present in the OFF lamina of the IPL but not in the ON lamina. (C) (Left) GABA_Aα1 receptor (yellow; signal within terminals) and protein kinase C (PKC; red) immunolabeling of RBC terminals in WT retina. The merged panel consists of the PKC signal and the receptor signal within PKC positive RBC terminals. (Right) Image of KO retina shows reduced GABA_Aα1 receptor immunofluorescence within RBC terminals. (Far right) Quantifications of receptor expression confirmed a significant reduction of GABA_Aα1 expression in the KO (mean ± sem = 0.8 ± 0.1) retina relative to WT (mean ± sem = 4 ± 0.1). (D) (Left) Immunolabeling of RBC terminals (PKC;red) in WT retina with antibodies

*Figure 1 continued on next page*

*Figure 1 continued*

against the ρ-subunit-containing GABA$_C$ receptor (GABA$_C$ρ; yellow – signal within RBC terminals). (Right) Image shows GABA$_C$ρ immunoreactivity within RBC terminals in the KO retina. (Far right) Quantification of RBC terminal GABA$_C$ρ receptor expression in KO (mean ± sem = 7.9 ± 0.6) retina relative to WT (mean ± sem = 8.4 ± 0.6). (E) Exemplar traces of evoked responses of an RBC after GABA puff application at its axon terminal. WT (Left, black trace); KO (Right, red traces). TPMPA (GABA$_C$ receptor antagonist) and GABAzine (GABA$_A$ receptor antagonist) were used to pharmacologically isolate GABA$_A$ and GABA$_C$ receptor-mediated components of the evoked responses. The GABA$_A$ component is revealed after application of TPMPA (labeled +TPMPA) and is eliminated upon the addition of GABAzine (TPMPA + GABAzine; labeled +GABAzine). Note the reduction of the GABA$_A$ receptor-mediated component in the KO relative to the WT. (F) Bar graph quantifying the GABA$_A$ and GABA$_C$-mediated component of RBC evoked responses in WT (black) and KO (red) retina. The mean ± sem peak amplitudes of GABA$_A$ + GABA$_C$ currents were 202.5 ± 5.5 pA in WT retina and 112.3 ± 5.6 pA in KO retina. The mean ± sem peak amplitudes of GABA$_A$ currents were 96.8 ± 5.2 pA in WT retina and 15.2 ± 1.5 pA in KO retina. The mean ± sem peak amplitudes of GABA$_C$ currents were 105.8 ± 1.8 pA in WT retina and 97 ± 4.3 pA in KO retina. Note that the significant reduction in the total response (GABA$_A$ + GABA$_C$) in the KO can be attributed to the reduction in the GABA$_A$-mediated component. In all figures, error bars indicate sem and 'n' refers to the number of cells analyzed except 1C, 1D, and *Figure 1—figure supplement 1E* occupancy quantifications where 'n' refers to the number of retinas analyzed.

The online version of this article includes the following figure supplement(s) for figure 1:

**Figure supplement 1.** GABA$_A$ inhibition remains intact in OFF cone bipolar cell (CBC), amacrine cell (AC), and ganglion cell (GC) processes in the knockout (KO) mouse.

plexiform layer (IPL) in the KO mice (*Figure 1B*), whereas the GABA$_A$α1 expression in the OFF sub-lamina remained unperturbed (*Figure 1—figure supplement 1A*). Co-labeling of GABA$_A$α1 with the RBC marker, protein kinase C (PKC), revealed near-complete absence of GABA$_A$α1 receptor subunit expression from RBC boutons in the KO retina (*Figure 1C*). This dramatic reduction in GABA$_A$α1 receptor expression in the KO retina was quantified by estimating the receptor percentage volume occupancy relative to the volume of the RBC axon terminal (see 'Materials and methods'). Despite the loss of GABA$_A$ receptors in KO RBC terminals, GABA$_C$ receptor expression was unchanged in KO RBC terminals (*Figure 1D*; quantifications of percentage GABA$_C$ volume occupancy). We confirmed the loss of GABA$_A$ receptors from KO ON BC axon terminals by measuring GABA-evoked currents from RBCs (*Figure 1E*). Puffing GABA on the axon terminals of RBCs elicited smaller currents in the KO retina. Upon application of a selective GABA$_C$ receptor blocker, TPMPA, we found that this decrease in total GABA-evoked current is due to a drastic reduction of the GABA$_A$ receptor-mediated current with unaltered GABA$_C$ currents in KO RBC terminals (*Figure 1F*). To confirm that GABA$_A$ receptors remain unaltered in the OFF sublamina, we performed whole-cell voltage clamp recordings of light evoked excitatory currents from OFF alpha transient GCs (a measure of the OFF BC output) which showed no differences (*Figure 1—figure supplement 1B*) in amplitude between KO and wildtype adult littermate control (WT) retina. This together with the unperturbed GABA$_A$α1 expression in the OFF sub-laminas confirms that GABAergic presynaptic inhibition across OFF BCs is not altered in the KO retina.

To eliminate the possibility of decreased GABA$_A$ receptor expression in AC and GC processes in the KO retina, we next determined expression of GABA$_A$ receptors across AC and GC processes that laminate in the same ON plexus of the retinal IPL as ON BC terminals (*Figure 1—figure supplement 1D–E*). We labeled AC and GC processes by labeling for the calcium binding protein calbindin that is specific to AC and GC processes in the IPL (*Haverkamp and Wässle, 2000*). To label all GABA$_A$ receptors across AC and GC processes we labeled for GABA$_A$β2/3 receptor subunits which are ubiquitously expressed across BC, AC, and GC processes in the IPL (*Greferath et al., 1995*). We could not label for specific GABA$_A$α receptor subunits in AC and GC processes as the composition of the GABA$_A$α receptor types across different AC and GC processes remains unknown and because our previous work has shown that GABA$_A$α1-containing receptors are enriched at BC processes (*Hoon et al., 2015*) but not GC processes (*Sawant et al., 2021*). Upon quantification of the percentage occupancy of GABA$_A$β2/3 receptors across ON-laminating AC and GC processes we observed comparable receptor amounts across genotypes (*Figure 1—figure supplement 1E*) confirming that expression of GABA$_A$ receptors is not impacted across AC and GC processes in the KO retina. Together, our findings demonstrate the targeted deletion of GABA$_A$ receptors only from the terminals of ON BCs in the KO retina.

## Increased sensitivity of ONα GCs to dim light stimuli in the KO retina

We chose the ONα GC retinal circuit as a means to explore the role of presynaptic inhibition in shaping retinal output across luminance (i) due to its well-characterized glutamatergic excitatory pathway – via RBC and Type 6 (T6) ON CBCs (*Grimes et al., 2014b*; *Schwartz et al., 2012*) – and (ii) because previous studies have shown that ON BCs, primarily RBC and T6 CBC, express *Grm6* in the *Grm6-tdTomato* transgenic mice (*Hoon et al., 2015*; *Kerschensteiner et al., 2009*) and are thus specifically targeted in our triple transgenic mouse line (Ai9/*Grm6*Cre/*Gabrg2 cKO*). This means that in the KO retina, the majority of the GABA$_A$ receptors lost from the IPL are from RBC and T6 CBC terminals (*Figure 1*; and see *Hoon et al., 2015*), thus providing a unique opportunity to study how GABA$_A$ receptor-mediated presynaptic inhibition alters the function of a specific retinal circuit, i.e. the ONα GC circuit (*Figure 1A*). We first probed the dim light sensitivity of ONα GC spike output using a full-field light stimulus that mostly activates the rods (*Figure 2*). The characteristic response of an ONα GC to a light step (0.5 s duration) has two distinct kinetic components – a fast transient phase and steady-state sustained phase – which results in action potential firing throughout the duration of the light stimulus (*Figure 2B*). Such a biphasic response to a sustained dim light step has previously been observed in the excitatory currents of the AII amacrine cell and reflects the intrinsic synaptic properties of the RBC to AII amacrine cell ribbon synapse (*Oesch and Diamond, 2011*, *Oesch and Diamond, 2019*). Furthermore, the transient component has been attributed to encode contrast whereas the sustained component has been shown to encode the absolute luminance (*Oesch and Diamond, 2011Oesch and Diamond, 2019*). In the KO retina, there is a sizeable increase in the spike response of ONα GCs for both its transient and sustained component (*Figure 2C and D*; see 'Materials and methods' for details). In addition to response amplitude, presynaptic inhibition is known to shape the kinetics of neuronal responses across diverse neural circuits (*Isaacson and Scanziani, 2011*; *Jadzinsky and Baccus, 2013*; *Ohliger-Frerking et al., 2003*). We analyzed the time course of the spike responses of the ONα GCs to the dim light step by estimating the time to peak of the response. There was no significant difference in the kinetics of the ONα GC spike responses between KO and WT retina under dim light conditions (*Figure 2E*).

To determine if the changes in the spike output are present in the excitatory synaptic inputs to the ONα GCs, we performed whole-cell voltage clamp recordings from the ONα GC and measured the excitatory synaptic current evoked by the above dim light stimuli (*Figure 2F*). The heightened response to the dim light stimuli is in fact more prominent in the excitatory synaptic currents of ONα GCs in the KO retina with nearly a twofold increase in the amplitude of the response in the KO retinas compared to that in the WT retinas (*Figure 2G*). Both the sustained and transient response components for the ONα GC are equally affected in the KO retina and hence the ratio of sustained to transient response remained unchanged between the WT and the KO retina for light evoked excitatory synaptic currents as well as for spike output (*Figure 2D and H*). We next estimated the kinetics of the ONα GC dim light evoked excitatory synaptic currents and found that the time to peak of the current response remained unchanged between the WT and the KO retina similar to that for the spike output (*Figure 2I*; see *Figure 2E*). To probe if the response differences in ONα GC between KO and WT retina are present across a broad range of dim light levels, we measured responses to brief light flashes (30 ms duration) of increasing intensity under dim light conditions (*Figure 2J–M*). Both the ONα GC spike output and the excitatory synaptic currents in the KO retina showed a marked increase over a considerable range of dim light flash intensities (*Figure 2K and M*). This indicates that light-triggered output from the RBC terminals is increased in absence of GABA$_A$ receptors across a wide range of dim light levels albeit not for the dimmest flashes. Moreover, enhancement of both the transient and sustained component of the ONα GC response to a sustained dim light stimulus indicates that perhaps both contrast and luminance encoding are altered at the dim light levels. Although the amplitude of the dim light evoked responses is perturbed, response kinetics remain unaltered in absence of GABA$_A$ receptor-mediated presynaptic inhibition. These results show that GABA$_A$ receptors at the RBC terminal play a key role in regulating the strength of dim light signals received by the ONα GCs presumably by controlling synaptic release from the RBC terminal. We cannot rule out a contribution of loss of GABA$_A$ receptors at the T6 ON CBC towards altered ONα GC sensitivity at dim light levels since rod-driven signals are routed from the AII amacrine via gap junctions to the ON CBC terminals and then onto the ONα GC.

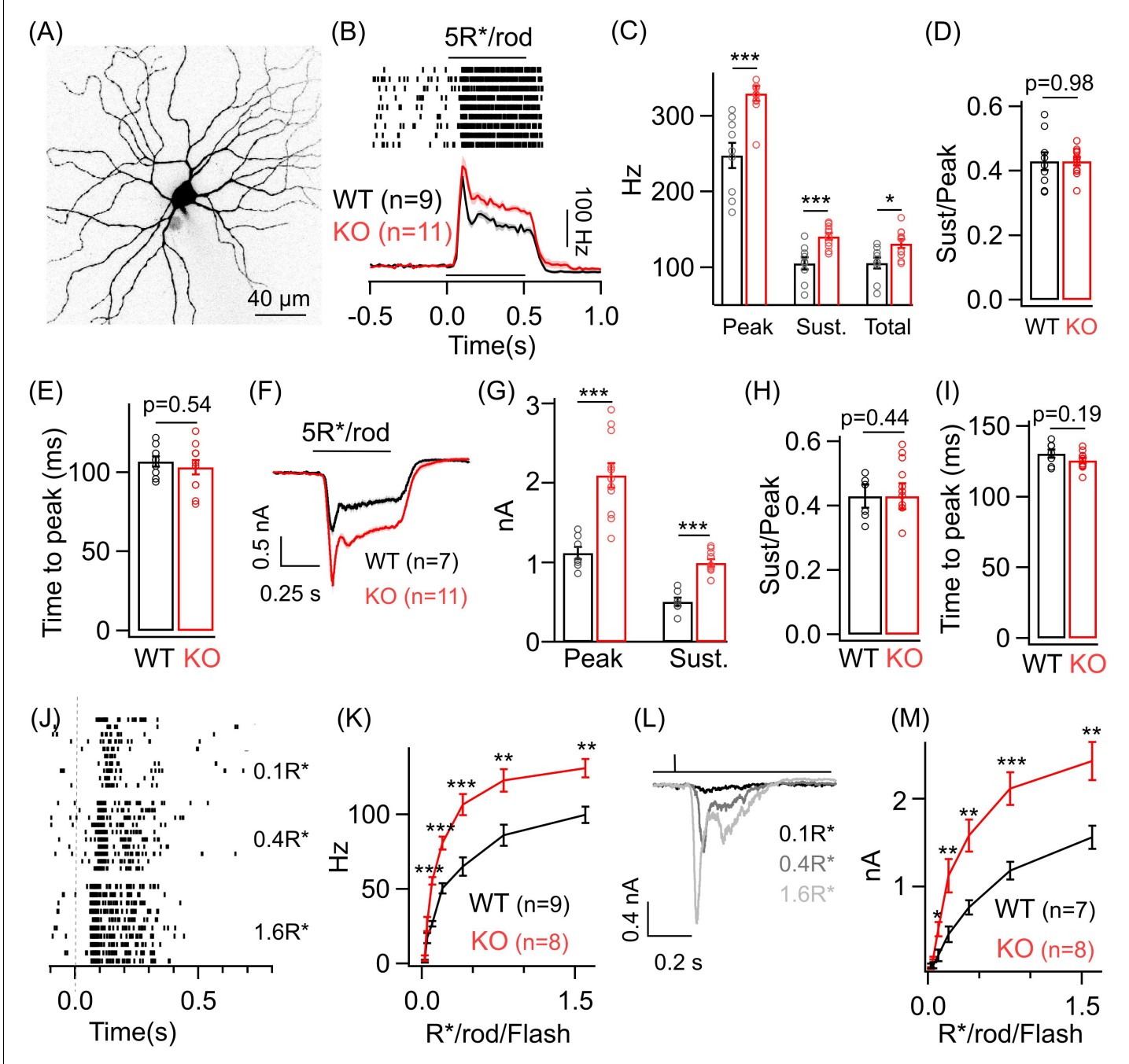

**Figure 2.** Dim light sensitivity of ONα ganglion cells (GCs) is perturbed without GABA_A presynaptic inhibition. (A) Exemplar image of an ONα GC filled with dye post-recording. (B) Raster plot showing an ONα GC spike response to a 0.5 s light step (that leads to five opsin photoisomerizations (R*) per rod photoreceptor) from darkness. Bottom panel shows average peri-stimulus time histograms (PSTH; binwidth of 20 ms) of the spike response from several ONα GCs in WT and KO mouse retina. Error bars (sem) shown in shaded colors henceforth for all average traces. The sample size for each experiment henceforth is mentioned next to the average traces and is the same for the following quantification represented in bar plots. (C) Bar plot comparing the peak (mean ± sem = 247.7 ± 16.8 Hz in WT and 329.7 ± 9.7 Hz in KO retina), sustained (mean ± sem = 105.4 ± 8.1 Hz in WT and 140.8 ± 4.4 Hz in KO retina) and total firing rate (mean ± sem = 105.6 ± 7.4 Hz in WT and 131.5 ± 5.7 Hz in KO retina) across ONα GCs as shown in (B) between WT and KO retina. (D) Bar plot comparing the ratio of sustained to peak firing rate of individual ONα GCs in WT (mean ± sem = 0.43 ± 0.03) and KO (mean ± sem = 0.43 ± 0.01) retina in response to light stimulus shown in (B). (E) Bar graph comparing the time to peak of spike PSTH (with a binwidth of 2 ms) across ONα GCs in WT (mean ± sem = 106.9 ± 3.2 ms) and KO (mean ± sem = 103.3 ± 4.5 ms) retina for the same data shown in (B). (F) Average excitatory synaptic currents measured across WT and KO ONα GCs elicited by the light stimulus described in (B). (G, H) Bar plot showing the light-evoked peak (mean ± sem = 1120.5 ± 78.8 pA in WT and 2096.2 ± 155.6 pA in KO retina) and sustained responses (mean ± sem = 505.2 ± 51.9

*Figure 2 continued on next page*

*Figure 2 continued*

pA in WT and 993.7 ± 45.5 pA in KO retina) and their ratio (mean ± sem = 0.45 ± 0.04 in WT and 0.49 ± 0.04 in KO retina) analyzed from individual ONα GCs. (I) Bar graph comparing the time to peak of the excitatory current response across ONα GCs in WT (mean ± sem = 130.5 ± 3 ms) and KO (mean ± sem = 125.8 ± 1.9 ms) retina for the same data shown in (F). (J) Spike trains from an exemplar ONα GC showing the response to brief (30 ms duration) light flashes that elicit 0.1, 0.4 and 1.6 R*/rod. (K) Peak spike rates of ONα GC in response to increasing flash strengths at dim light levels in WT and KO retinas. (L) Excitatory synaptic currents measured from an exemplar WT ONα GC elicited by light flashes shown in (J). (M) Peak excitatory current response of ONα GCs in response to increasing flash strengths at dim light levels in WT and KO retinas.

## ONα GCs in the KO retina exhibit changes in amplitude and kinetics of responses to light stimuli that preferentially excite the cone photoreceptors

Given the increase in dim light sensitivity of ONα GCs in the KO retina, we wanted to test if this persists even for brighter light levels that primarily activate the cone pathway. To do so, we adapted the retina to a background luminance of ~1000 R*/cone/s, which mostly saturates the rods (*Grimes et al., 2018*) and allows us to preferentially probe the cone-mediated ONα GC responses. We first measured the spike response of ONα GCs to a full-field 100% contrast increment (*Figure 3A*). The ONα GC response at these cone-dominated light levels also shows two kinetic phases, i.e. a transient and sustained phase (*Figure 3B*), similar to the responses at dim light levels shown in *Figure 2*. We observed a nearly twofold increase in the peak firing rate of the ONα GC in the KO retina compared to that in the WT retina. However, we did not see a systematic difference in the sustained phase of the spike response of ONα GCs between WT and KO retina (*Figure 3—figure supplement 1B*). Next, we compared the kinetics of the spike responses of the ONα GCs between KO and WT retina (*Figure 3D*). Interestingly, the time to peak of the ONα GC spike responses in the KO retina were significantly slower than in the WT retina in contrast to our earlier observations under dim light conditions (*Figure 3D*). To determine if the changes in the amplitude and kinetics of the cone-mediated spike responses are reflected in the excitatory inputs, we performed whole-cell voltage clamp recordings from ONα GCs in KO and WT retina in response to the above stimuli (*Figure 3E*). Both the transient and sustained phase of the light-evoked excitatory synaptic current were ~2-fold larger in the KO retina compared to that in the WT retina (*Figure 3F*). The ratio of the sustained-to-transient phase of the ONα GC response remained unchanged between KO and WT retina (*Figure 3G*). Upon comparing the response kinetics of ONα GC excitatory currents between KO and WT retina, we observed that the time to peak was higher in the KO retina than in the WT retina similar to the effects on the spike output seen earlier (*Figure 3H*). We further quantified the time of onset of the sustained phase of the light-evoked excitatory currents in ONα GC but did not observe any significant changes between KO and WT retina (*Figure 3—figure supplement 1D,E*). The increase in response amplitude and slower time course of the ONα GC excitatory synaptic currents in the KO retina was also evident when we presented a briefer light flash of 10 ms duration (*Figure 3I–L*). In addition to the longer time taken to reach peak response (*Figure 3K*), we also found that the decay time of the flash-evoked ONα GC excitatory synaptic currents (from the peak to the baseline) was significantly longer in KO compared to WT retina (*Figure 3L*). This suggests that $GABA_A$ receptor-mediated presynaptic inhibition regulates both the activation and recovery of the cone-mediated signals at the BC to ONα GC synapse. Given that the size of the flash-evoked ONα GC excitatory currents is ~2-fold larger in KO retina, we wanted to further probe the response recovery, particularly the overshoot after the flash response reaches the baseline (*Figure 3M*). We estimated the amplitude of the response overshoot from the baseline and observed no difference between ONα GCs in KO and WT retina (*Figure 3M*).

To ensure that postsynaptic inhibition acting directly on the ONα GC is not significantly perturbed in the KO retina and the differences we see at the level of spike output are due to changes in the excitatory input, we compared light-evoked inhibitory currents from ONα GC to a 100% contrast increment step (*Figure 3—figure supplement 2A*). There was no change in light-evoked postsynaptic inhibition in ONα GC in the KO retina compared to that in the WT retina indicating that increase in the excitatory inputs most likely cause the increase in the spike output of ONα GC light responses in the KO retina (*Figure 3—figure supplement 2B*).

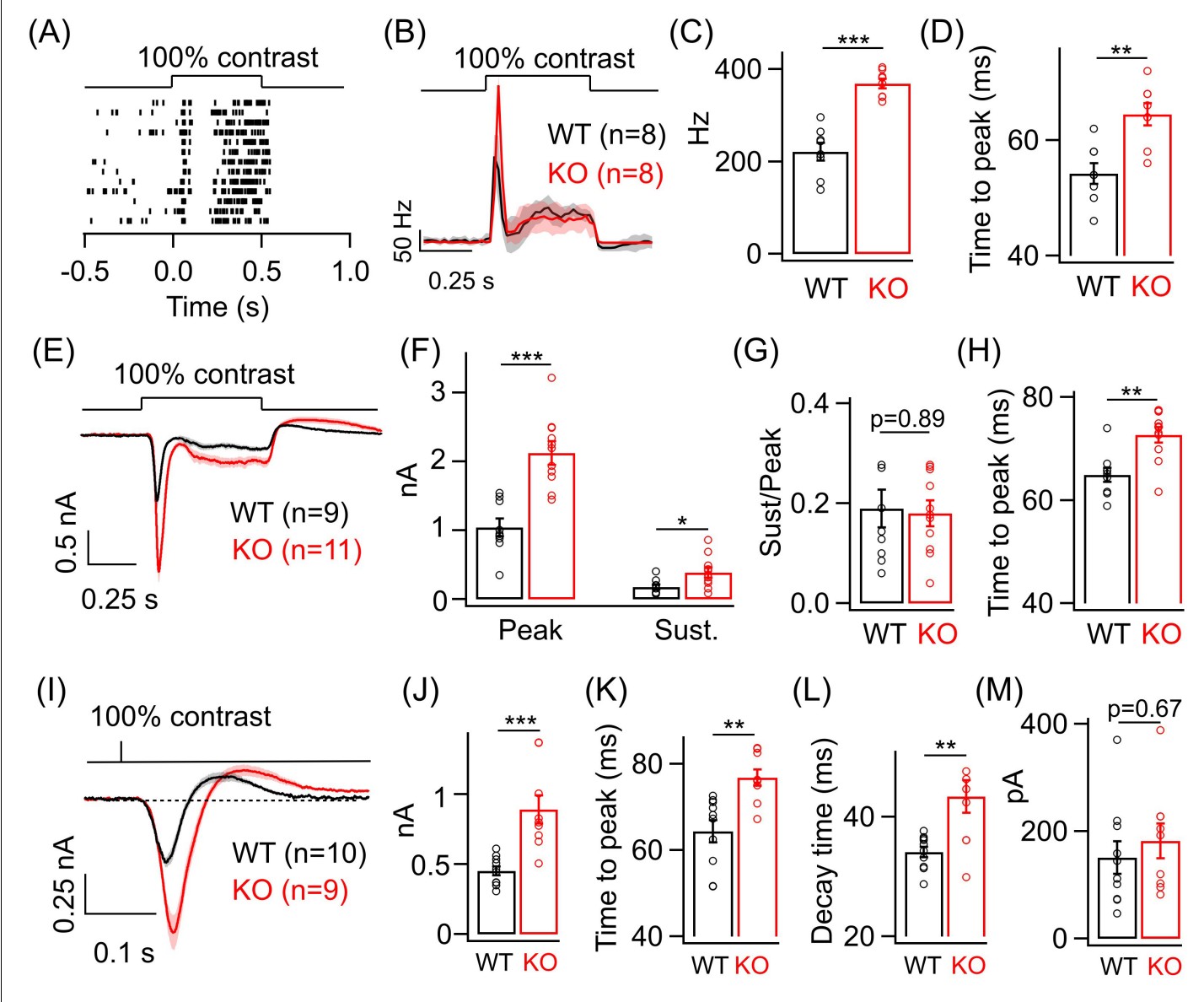

**Figure 3.** Lack of GABA_A presynaptic inhibition alters ONα ganglion cell (GC) light responses at cone light levels. (**A**) Exemplar spike raster from an ONα GC in WT retina in response to a 100% contrast step from a background luminance of ~1000R*/S cone/s, where cones dominate retinal responses. (**B**) Average PSTH (binwidth of 20 ms) of spike response to the light step in A across several ONα GCs with spike rate expressed in Hz (spikes/s). (**C**) Bar plot showing quantification of the peak firing rates across ONα GCs in WT (mean ± sem = 221.2 ± 18.9 Hz) and KO retina (mean ± sem = 368.5 ± 10.2 Hz). (**D**) Bar graph comparing the time to peak of spike PSTH (with a binwidth of 2 ms) across ONα GCs in WT (mean ± sem = 54.3 ± 1.8 ms) and KO (mean ± sem = 64.5 ± 1.9 ms) retina for the same data shown in (**B**). (**E**) Average excitatory synaptic current elicited in response to a 100% contrast step across ONα GCs in WT and KO retina. (**F**) Quantification of peak and sustained current amplitudes in response to the 100% contrast step in E. The mean ± sem peak amplitudes were 1042.3 ± 127.9 pA in WT retina and 2126 ± 169.5 pA in KO retina. The mean ± sem amplitudes of the sustained phase were 176.7 ± 36.2 pA in WT retina and 386.9 ± 75.5 pA in KO retina. (**G**) Quantification of ratio of sustained to peak amplitude in F. The mean ± sem ratios were 0.19 ± 0.04 in WT retina and 0.18 ± 0.03 in KO retina. (**H**) Bar graph comparing the time to peak of the excitatory current response across ONα GCs in WT (mean ± sem = 64.9 ± 1.4 ms) and KO (mean ± sem = 72.7 ± 1.5 ms) retina for the same data shown in (**E-G**). (**I**) Average excitatory synaptic currents in response to 10 ms flash of 100% contrast across ONα GCs in WT and KO retina. (**J**) Quantification of peak current amplitude in response to the 10 ms flash of 100% contrast step in WT (mean ± sem = 451.6 ± 32.2 pA) and KO (mean ± sem = 890.9 ± 100.8 pA) retina as shown in I. (**K**) Bar graph comparing the time to peak of the excitatory current response across ONα GCs in WT (mean ± sem = 64.4 ± 2.6 ms) and KO (mean ± sem = 76.8 ± 1.9 ms) retina for the same data shown in (**I**). (**L**) Quantification of decay time of the excitatory current response, i.e. time for the response in (**I**) to return from the peak to the baseline shown in dotted line, across ONα GCs in WT (mean ± sem = 34.1 ± 0.9 ms) and KO (mean ± sem = 43.4 ± 2.8 ms) retina for the same data shown in (**I**). (**M**) Quantification of the rebound amplitude

*Figure 3 continued on next page*

**Figure 3 continued**

of the excitatory current response across ONα GCs in WT (mean ± sem = 151 ± 30.4 pA) and KO (mean ± sem = 182.4 ± 32.2 pA) retina for the same data shown in (I).

The online version of this article includes the following figure supplement(s) for figure 3:

**Figure supplement 1.** Analysis of the sustained phase of the light-evoked spike and excitatory current response in ONα ganglion cells (GCs).

**Figure supplement 2.** Post-synaptic inhibition remains unchanged in ONα ganglion cell (GC) in the knockout (KO) retina.

These results suggest that loss of GABA$_A$ receptor-mediated presynaptic inhibition alters the amplitude of both rod- and cone-driven signals but only the time course of cone-driven signals received by the ONα GCs.

## Temporal sensitivity and contrast encoding of ONα GCs are altered in the KO retina

To compare the temporal filtering and contrast sensitivity of the ONα GCs in WT and KO retina, we used a random time-varying stimulus consisting of a range of temporal frequencies and contrasts (*Figure 4A*; see 'Materials and methods'). To characterize the responses, we used a linear-nonlinear (LN) model that provides a relatively simple description of how light inputs are transformed into neuronal responses and provides an effective way of determining contrast-dependent changes in the amplitude and kinetics of the light response of retinal neurons (*Beaudoin et al., 2007*; *Kim and Rieke, 2001*; *Sinha et al., 2017*). The model has two components – a linear filter that describes the time course of the neuronal response and a time-invariant or 'static' nonlinearity that transforms the filtered stimulus into neuronal responses (*Beaudoin et al., 2007*; *Kim and Rieke, 2001*; *Rieke, 2001*). We focused on excitatory synaptic currents since the loss of GABA$_A$ receptors at ON BC terminal in the KO retina will directly impact the ON BC output and hence the glutamatergic synaptic input onto the ONα GCs. We measured ONα GC excitatory synaptic currents in response to the time-varying stimuli that were modulated at two background light levels – one that preferentially activate rods and the other that selectively excites the cone photoreceptors (*Figure 4A,H*). At cone light levels, linear filters show that the time course of the ONα GC excitatory current response in the KO retina is considerably slower than in the WT retina (*Figure 4B*). Both the time to peak and the decay time of the linear filters were significantly longer for ONα GC in KO retina compared to WT retina (*Figure 4C*) similar to the above results from the responses to brief light flashes (*Figure 3K,L*). This suggests that lack of GABA$_A$ receptor-mediated presynaptic inhibition alters the temporal filtering of the T6 CBC output and hence the excitatory inputs in the ONα GCs by likely attenuating higher frequencies more and lower frequencies less. We next compared the static nonlinearity of the ONα GCs in WT and KO retina (*Figure 4E*). We first quantified the response range which we defined as the absolute difference between the maximum and the minimum value of the measured current response to the chosen contrast range (*Figure 4F*). This response range, i.e. dynamic range of the excitatory currents for the given range of contrasts, was ~2-fold larger for the ONα GCs in the KO retina in comparison to their counterparts in the WT retina (*Figure 4F*). This is consistent with our above results of the ONα GC responses to light flash/step of fixed intensity (*Figure 3*). Given the sizeable change in the response amplitude and the response range, we assessed if the contrast gain is altered in the ONα GCs in KO retina compared to that in WT retina. This can be estimated from the slope of the nonlinearity or the height of the linear filter since both the linear filter and the static nonlinearity share contrast-dependent changes in the amplitude of the neuronal response (*Kim and Rieke, 2001*; *Rieke, 2001*). To unambiguously measure contrast gain, we normalized the linear filter and then compared the slope of the nonlinearity between ONα GC responses in KO and WT retina (see 'Materials and methods'). The nonlinearities of the ONα GC response in KO retina had a steeper slope and upon quantification the slope differed by a factor of ~2 compared to the ONα GC nonlinearities in WT retina (*Figure 4G*). These results suggest that GABA$_A$ presynaptic inhibition tightly regulates contrast sensitivity of the ONα GC excitatory inputs and restricts the response size used for encoding over a range of contrast.

We repeated the above experiments on ONα GCs in WT and KO retina under a dim light background (*Figure 4H–M*). The nonlinearity of the ONα GC in the KO retina also had a bigger response range than in WT retina similar to that observed under cone-driven light levels (*Figure 4K,L*). The

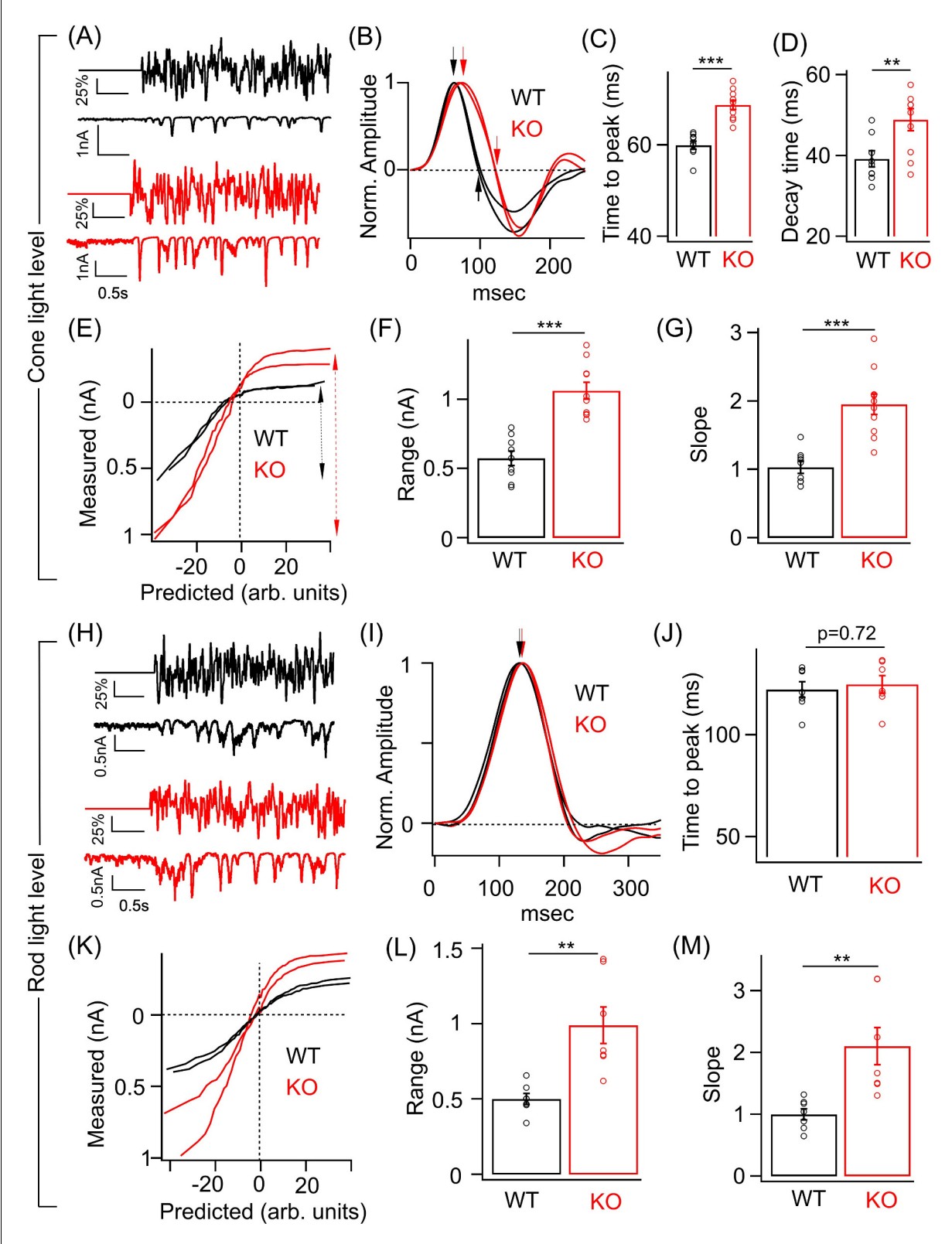

**Figure 4.** Perturbed ONα ganglion cell (GC) responses to time-varying light stimuli in absence of GABA$_A$ presynaptic inhibition. (**A**) (Top) Excerpt of the time-varying random white noise stimulus presented at a background luminance of 1000 R*/S cone/s. (Bottom) The resulting excitatory synaptic response used to derive the linear filter and static nonlinearity that relate the stimulus to the response. (**B**) Exemplar time-reversed linear filters for the responses to noise stimuli for two ONα GCs from WT and KO retina. The black and red arrows point to the time to peak and the time point of decay to

*Figure 4 continued on next page*

*Figure 4 continued*

the baseline. Quantification of the time to peak (C) and the decay time (D) in linear filters for responses to stimuli (cone light levels) across ONα GCs in WT (n = 9 cells) and KO (n = 10 cells) retina. The mean ± sem peak times to peak of the linear filters were 59.9 ± 0.9 ms in WT retina and 68.5 ± 0.9 ms in KO retina. The mean ± sem decay times of the linear filters were 39.2 ± 2 ms in WT retina and 48.9 ± 2.7 ms in KO retina. (E) Exemplar static nonlinearities of two ONα GCs from KO and WT retina for the noise stimuli. (F) Quantification of the response range (denoted by the dotted black and red arrows) in (E) across ONα GCs in WT (n = 9 cells; mean ± sem = 573.3 ± 52 pA) and KO (n = 10 cells; mean ± sem = 1063.5 ± 61.4 pA) retina. (G) Quantification of the nonlinearity slope (see 'Materials and methods') across ONα GCs in WT (n = 9 cells; mean ± sem = 1 ± 0.1) and KO (n = 10 cells; mean ± sem = 2 ± 0.2) retina. (H) (Top) Excerpt of the time-varying stimulus presented at a dim background luminance of 10 R*/rod/s. (Bottom) The resulting excitatory synaptic response used to derive the linear filter and static nonlinearity that relate the stimulus to the response. (I) Exemplar time-reversed linear filters for the responses to noise stimuli (under rod light levels) for two ONα GCs from WT and KO retina. The black and red arrows point to the time to peak. (J) Quantification of the time to peak in linear filters for responses to noise stimuli (rod light levels) across ONα GCs in WT (n = 7 cells; mean ± sem = 122.2 ± 3.9 ms) and KO (n = 7 cells; mean ± sem = 124.8 ± 4.3 ms) retina. (K) Exemplar static nonlinearities of ONα GC responses for the noise stimuli from WT and KO retina. (L) Quantification of the response range across ONα GCs in WT (n = 7 cells; mean ± sem = 500.6 ± 37.1 pA) and KO (n = 7 cells; mean ± sem = 990 ± 121.9 pA) retina. (M) Quantification of the nonlinearity slope (see 'Materials and methods') across ONα GCs in WT (n = 7 cells; mean ± sem = 1 ± 0.1) and KO (n = 7 cells; mean ± sem = 2.1 ± 0.3) retina.

slope of the nonlinearity, i.e. contrast gain was also ~2-fold higher for the ONα GC in the KO retina than that in the WT retina (*Figure 4M*). However, no significant difference in the kinetics of the linear filters between the genotypes was observed (*Figure 4J*). Our findings of alterations in response size and time course using randomly flickering stimuli are consistent with the results obtained above (*Figures 2* and *3*) using fixed intensity stimuli. Thus, loss of GABA$_A$ receptor-mediated presynaptic inhibition at the RBC and T6 CBC terminal alters the contrast sensitivity and kinetics of the ONα GC excitatory inputs. Importantly the changes in response amplitude and kinetics observed at the level of ONα GC excitatory inputs in the KO retina are reflected in the spike output which highlights the importance of GABA$_A$ presynaptic inhibition as a mechanism in shaping visual signals being transmitted out of the retina.

## Discussion

Presynaptic inhibition is an important mechanism for regulating a neuron's input-output relationship. However, it has been difficult to isolate its precise contribution in most retinal circuits due to lack of receptor type-, cell type-, and circuit-specific perturbations. Here we have taken advantage of a previously used (*Hoon et al., 2015*) transgenic manipulation in mouse retina that selectively eliminates a specific population of inhibitory receptors – GABA$_A$ receptors – from the axon terminals of defined types of presynaptic neurons – RBC and T6 CBCs – and determined its impact on the light-evoked response properties of one of the well characterized downstream retinal output neuron – the ONα GC. We show that GABA$_A$ receptor-mediated presynaptic inhibition is crucial for regulating the amplitude and contrast sensitivity of both rod and cone-driven signals routed to the ONα GCs. Interestingly, GABA$_A$ presynaptic inhibition shapes the kinetics of only cone-driven signals but not rod-driven signals reaching the ONα GCs. We show that the impact on the gain and kinetics of visual signals as observed in the excitatory synaptic inputs received by the ONα GCs, is propagated to its spike output. Thus, GABA$_A$ presynaptic inhibition is a key mechanism of gain control and temporal filtering for the ONα GC retinal circuit.

### GABA$_A$ presynaptic inhibition shapes rod and cone signaling in the ONα GC retinal circuit

Several studies including our current study have shown that GABA$_A$ and GABA$_C$ receptors in ON BC terminals contribute to nearly equal amplitudes of GABA-evoked currents (*Hoon et al., 2015*; *Sinha et al., 2020*). However, most of what we know about the role of presynaptic inhibition in shaping the retinal RBC (*Pan et al., 2016*) and ON CBC synaptic output has been attributed to GABA$_C$ receptor-mediated inhibition (*Oesch and Diamond, 2019*; *Sagdullaev et al., 2006*). In the current study, we show that GABA$_A$ receptor-mediated presynaptic inhibition plays an equally important role in regulating the dynamic range and contrast sensitivity as GABA$_C$ receptor-mediated presynaptic inhibition for both the RBC output under dim light conditions as well as for cone-mediated signals via the ON CBC synapse. GABA$_A$ receptor-mediated presynaptic inhibition restricts the response range and contrast gain of ONα GC responses which could allow the ONα GC retinal circuit to

encode over a wider range of contrast and luminance without being saturated. This is a common feature of light adaptation whereby retinal neurons match their neural gain to the prevailing visual inputs such that they can continue to efficiently signal over a broad range of light inputs (*Rieke and Rudd, 2009*). Thus, our results together with previous findings (*Oesch and Diamond, 2019*; *Sagdullaev et al., 2006*) suggest that both GABA$_A$ and GABA$_C$ presynaptic inhibition play a key role in regulating response amplitude and contrast encoding under rod- and cone-dominant lighting conditions.

A central role of presynaptic inhibition that has not been extensively explored in the retina is how it shapes the temporal sensitivity of rod- and cone-driven signals routed via specific neural circuits (*Asari and Meister, 2012*). Temporal processing is crucial to encode dynamic features of visual signals such as motion (*Jadzinsky and Baccus, 2013*). Temporal filtering, i.e. sensitivity to certain temporal patterns, is different across RGC types, and synaptic inhibition is a common mechanism that is known to shape temporal filtering in most neural circuits (*Baden et al., 2016*). Presynaptic inhibition is well positioned to decrease synapse output and attenuate steady inputs thus temporally filtering signals received by RGCs. Signals originating in rod vs. cone photoreceptors are known to exhibit remarkably distinct temporal characteristics with rod signals being substantially slower compared to cone signals (*Cangiano et al., 2012*; *Ingram et al., 2016*). This difference is reflected in our results from both fixed intensity and randomly modulating stimuli where the time course of rod signals was nearly 2-fold slower than the cone-mediated signals in WT retina measured at the level of ONα GCs (*Figure 2E and I* vs. *Figure 3D,H and K*; *Figure 4C vs J*). Our findings show that GABA$_A$ presynaptic inhibition speeds up the time course of cone-mediated signals but not rod-driven signals in the ONα GC retinal circuit. This could be because under dim light conditions where photons are sparse, a longer integration time of the rod signals by the downstream circuit may benefit signal detection (*Field et al., 2005*). In this case, temporal filtering by mechanisms such as presynaptic inhibition could be detrimental to the detection of sparse signals such as single photons. In fact, recent studies have shown that ONα GCs are one of the most sensitive GC types in the mouse retina under dim light conditions and comprise the major conduit for relaying single photon signals out of the retina (*Smeds et al., 2019*). Hence, minimizing temporal filtering by GABA$_A$ presynaptic inhibition may help prolong the duration of signal integration and may improve sensitivity of the ONα GCs for single photon signaling in near complete darkness.

Besides signaling efficiently at absolute threshold in darkness, the ONα GC retinal circuit also integrates cone-driven signals and can operate under high luminance conditions (*Grimes et al., 2014b*; *Schwartz et al., 2012*; *Sonoda et al., 2018*). Our results show that GABA$_A$ presynaptic inhibition limits the response size and contrast gain while speeding up cone-driven signals reaching the ONα GCs. This can have two potential advantages. First, encoding contrast with a smaller response amplitude might allow to effectively signal over a broader dynamic range of contrasts without being saturated. As the contrast range explored in our experiments represent a small fraction of the contrast distribution present in natural scenes, such a gain control mechanism would help match the contrast sensitivity of the ONα GC retinal circuit to the statistics of the prevailing light inputs. Second, being able to signal fast changes in light inputs might aid the ONα GC retinal circuit in the efficient encoding of dynamic features such as during motion.

Our findings that GABA$_A$ presynaptic inhibition regulates the gain and kinetics of visual signals in the ONα GC retinal circuit is consistent with previous studies in other neural circuits, besides the retina, where presynaptic inhibition has been shown to play a central role in gain control and temporal filtering of neural signals (*Baden and Hedwig, 2010*; *Chen and Regehr, 2003*; *Fink et al., 2014*; *Frerking and Ohliger-Frerking, 2006*). For instance, presynaptic inhibition mediated by GABAergic interneurons contributes to motor behavior in the spinal cord, where it controls the gain of sensory afferents and mediates smooth muscle movement (*Fink et al., 2014*). In the olfactory system, GABAergic presynaptic inhibition of the olfactory sensory axon terminals serves as a primary gain control mechanism to maintain odor sensitivity over a wide range of inputs (*Olsen and Wilson, 2008*; *Root et al., 2008*). Additionally, presynaptic inhibition mediated by GABA receptors controls temporal contrast enhancement and modifies odor-guided navigation in *Drosophila melanogaster* (*Raccuglia et al., 2016*).

## Presynaptic inhibition regulates neurotransmitter release and synapse arrangement

The role of presynaptic inhibition is particularly important for the ONα GC pathway. Previous studies have shown that excitatory synaptic inputs dictate the spike output of the ONα GCs (*Murphy and Rieke, 2006*), and we show that GABA$_A$ presynaptic inhibition is a key mechanism well-poised to shape the ONα GC excitatory synaptic inputs. Both GABA$_A$ and GABA$_C$ receptors are localized at axon terminals of ON BCs, but they have been shown to be present at spatially distinct sites at RBC terminals relative to the site of synaptic release, i.e. ribbon (*Grimes et al., 2015*). Pharmacological blockade and genetic deletion of GABA$_C$ receptors have shown that GABA$_C$ receptor-mediated presynaptic inhibition regulates the extent of multivesicular glutamate release at the bipolar ribbon-type synapses (*Oesch and Diamond, 2019*; *Sagdullaev et al., 2006*). Particularly in the ON CBC synapse, loss of GABA$_C$ receptors results in activation of the perisynaptic NMDA receptors on RGC dendrites by glutamate spillover from the synapse thus enhancing synaptic output (*Sagdullaev et al., 2006*). This could be a potential underlying mechanism for the enhanced light-evoked response we observe in ONα GCs in absence of GABA$_A$ receptors on the ON CBC terminals.

In RBCs, luminance and contrast are encoded via dynamic release and replenishment of the readily releasable pool (RRP) of synaptic vesicles located at the ribbon (*Oesch and Diamond, 2011*; *Oesch and Diamond, 2019*). A step increase in luminance results in contrast encoding via a transient bout of vesicle release from the RRP, which corresponds to a transient peak in the AII AC excitatory current. The size of the remaining vesicle pool is used to encode luminance and corresponds to the sustained component of the excitatory postsynaptic current. A17 AC-mediated feedback inhibition on RBCs acting via GABA$_C$ receptors has been implicated in regulating the extent of RRP depletion and synaptic output from RBCs which in turn shapes luminance and contrast encoding across a range of dim light levels (*Oesch and Diamond, 2019*). Our results show that lack of GABA$_A$ receptors in the RBC and T6 CBC terminals affects both the transient and sustained components of RBC and T6 CBC output as measured from the impact on the ONα GC excitatory inputs under rod- and cone-dominant light conditions. This indicates that both contrast and luminance encoding in the RBC and T6 CBC pathway might be shaped by GABA$_A$ receptor-mediated presynaptic inhibition. However, we cannot distinguish between the contribution of 'feedback' vs. 'lateral' GABA$_A$ receptor-mediated presynaptic inhibition at RBC and T6 CBC terminal on ONα GC function. Feedback presynaptic inhibition on BC terminals is mediated by an AC that is activated by the same BC it provides inhibition onto, and lateral presynaptic inhibition is mediated by ACs activated by other BCs (*Asari and Meister, 2012*; *Grimes et al., 2015*).

It is well established that for most synapses the input-output relationship of membrane voltage vs. transmitter release is nonlinear and has a sigmoidal shape (*Neher and Sakaba, 2008*). Previous studies have in fact proposed that this synaptic transfer function of the ON CBC terminals has a sigmoidal relationship between membrane voltage and glutamate release with a steep nonlinear foot (*Grimes et al., 2014b*). The ON CBC or presynaptic membrane potential can be influenced by gap junctional coupling from the AII amacrine cell processes which can alter synapse output by changing its location on the voltage-release curve. This form of regulating synaptic output has been shown to play a critical role in shaping ONα GC function (*Grimes et al., 2014b*). GABA$_A$ receptor-mediated presynaptic inhibition could also regulate the presynaptic membrane voltage of the ON CBC, thereby controlling the set point of the synapse input-output curve and hence the ONα GC excitatory inputs.

ON BC synapses are specialized ribbon synapses that often have defined postsynaptic partners. One such example is the RBC output synapse onto AII and A17 AC processes (*Grimes et al., 2015*). Our recent study showed that presynaptic inhibition plays a key role in the precise assembly of the ribbon synapse at the RBC terminal and organization with correct postsynaptic partners (*Sinha et al., 2020*). Interestingly, under conditions where expression of both GABA$_A$ and GABA$_C$ receptors in the RBC terminals are downregulated, such as during lack of global inhibitory transmitter release or loss of specific synaptic adhesion molecules, ultrastructural analysis revealed that the RBC ribbon synapse is misorganised and makes erroneous connections with postsynaptic partners (*Sinha et al., 2020*). Therefore, despite the downregulation of both GABA$_A$ and GABA$_C$ receptors at the RBC terminal in this situation, there is decreased dim light sensitivity of the ONα GC output

(*Sinha et al., 2020*) probably due to a reduced feedforward excitatory drive as a result of synaptic mis-arrangements at the RBC terminal. In the KO mice used in the current study, there is a drastic reduction of GABA$_A$ receptor expression but unaltered GABA$_C$ receptor expression in the RBC terminals. Given that dim light sensitivity is increased in the KO retina, it will be interesting in the future to use ultrastructural techniques in the KO retina to determine if the selective reduction of GABA$_A$ receptors results in any organizational deficits of RBC output (ribbon) synapse assembly.

In conclusion, our study provides the first characterization of how selective perturbation of GABA$_A$ receptor-mediated inhibition at ON BC terminals impacts visual signaling of a well-characterized GC circuit. Future studies will be needed to explore how presynaptic inhibition regulates functional properties of other ganglion cell pathways as well as its contribution to shaping the receptive field organization of ganglion cell types.

# Materials and methods

### Key resources table

| Reagent type (species) or resource | Designation | Source or reference | Identifiers | Additional information |
|---|---|---|---|---|
| Genetic reagent (*Mus musculus*) | *Gabrg2* | Jackson Laboratory | JAX Stock# 016830 RRID:IMSR_JAX:016830 | Transgenic mouse; floxed mice with *loxP* sites flanking *Gabrg2* |
| Genetic reagent (*Mus musculus*) | Ai9 | Jackson Laboratory | JAX Stock# 007909 RRID:IMSR_JAX:00790 | Transgenic mouse; cre-dependent tdTomato expression |
| Genetic reagent (*Mus musculus*) | *Grm6*-Cre | Rachel Wong (*Hoon et al., 2015*) | N/A | Transgenic mouse; cre-driver line |
| Antibody | Anti-PKC clone MC5 (mouse monoclonal) | Sigma | Catalog # P5704; RRID:AB_477375 | (1:1000) |
| Antibody | Anti-GABA$_A$α1 (guinea pig polyclonal) | *Fritschy and Mohler, 1995* | Generated in Jean-Marc Fritschy's Lab | (1:5000) |
| Antibody | Anti-GABA$_C$ (rabbit polyclonal) | *Enz et al., 1996* | Generated in Heinz Wässle and Joachim Bormann's Lab. | (1:500) |
| Antibody | Anti-Dsred (rabbit polyclonal) | Clontech | | (1:1000) |
| Antibody | Anti-synaptotagmin2 (mouse monoclonal) | Zebrafish International Resource center | Cat# znp-1; RRID:AB_10013783 | (1:1000) |
| Antibody | Anti-calbindin (rabbit polyclonal) | Swant Inc. | Swant Cat# CB38; RRID:AB_10000340 | (1:1000) |
| Antibody | Anti-GABA$_A$β2/3, (mouse monoclonal) | MilliporeSigma | Cat# MAB341; | (1:500) |
| Chemical compound, drug | Ames | Sigma | A1420 | |
| Chemical compound, drug | Alexa 594 | Thermofisher | A10442 | |
| Chemical compound, drug | Vectashield | Vector Labs | Cat# H-1000, RRID:AB_2336789 | |
| Chemical compound, drug | GABAzine (SR-95531) | Sigma | S106 | |

*Continued on next page*

*Continued*

| Reagent type (species) or resource | Designation | Source or reference | Identifiers | Additional information |
|---|---|---|---|---|
| Chemical compound, drug | GABA | Sigma | A2129 | |
| Chemical compound, drug | TPMPA | Tocris | 1040 | |
| Software, algorithm | Symphony | https://github.com/symphony-das | | |
| Software, algorithm | ScanImage | http://scanimage.vidriotechnologies.com/ PMID:12801419 | RRID:SCR_014307 | |
| Software, algorithm | MATLAB | http://www.mathworks.com/products/matlab/ | RRID:SCR_001622 | |
| Software, algorithm | IGOR Pro | https://www.wavemetrics.com/ | RRID:SCR_000325 | |
| Software, algorithm | Amira | https://www.thermofisher.com/global/en/home/industrial/electron-microscopy/electron-microscopy-instruments-workflow-solutions/3d-visualization-analysis-software/amira-life-sciences-biomedical.html | RRID:SCR_007353 | |
| Software, algorithm | ImageJ | https://ImageJ.net | RRID:SCR_003070 | |

## Animal handling and ethic statement

All experiments and animal care were conducted in accordance with the Institutional Animal Care and Use Committee (IACUC) of the University of Wisconsin-Madison and the National Institutes of Health. Animals were housed in a 12 hr light/dark cycle. Ai9/*Grm6*Cre/*Gabrg2* cKO and littermate control adult (2–4 months) mice of both sexes were used in this study. The Ai9/*Grm6*Cre/*Gabrg2* triple transgenic mouse line was chosen because it allowed for selective perturbation of inhibitory receptor GABA$_A$ expression specifically in ON (RBC and T6 CBCs) BCs by genetic deletion of *Gabrg2* (GABA$_A$ receptor, subunit gamma 2) in these cells (*Hoon et al., 2015*). Loss of *Gabrg2* causes reduced presence of axonal but not dendritic GABA$_A$α1 receptors in T6 CBCs (*Hoon et al., 2015*). The triple transgenic was created by crossing *Gabrg2* floxed mutant mice (Jackson Laboratory, RRID:IMSR_JAX:016830) (*Schweizer et al., 2003*) with a transgenic mouse line *Grm6*–Cre in which Cre-recombinase is expressed by ON BCs shortly after their differentiation (*Hoon et al., 2015*; *Kerschensteiner et al., 2009*; *Morgan et al., 2006*). In order to label Cre-expressing cells with the red fluorescent protein tdTomato, the *Gabrg2* floxed/*Grm6*-Cre mice were further crossed into the Ai9 reporter line (Jackson Laboratory, RRID:IMSR_JAX:00790).

## Immunohistochemical labeling

Retinas were isolated in cold oxygenated mouse artificial cerebrospinal fluid (mACSF, pH 7.4, 119 mM NaCl, 2.5 mM KCl, 2.5 mM CaCl$_2$, 1.3 mM MgCl$_2$, 1 mM NaH$_2$PO$_4$, 11 mM glucose, and 20 mM HEPES). Retinas were flattened onto filter paper (Millipore, HABP013) and fixed for 15 mins in 4% (wt/vol) paraformaldehyde prepared in mACSF. Retinas were rinsed in phosphate buffer (PBS) and then incubated in a blocking solution (5% donkey serum and 0.5% Triton X-100). The retinas were next incubated with primary antibody over 3 nights at 4℃. Primary antibodies used were anti-PKC (1:1000, mouse, Sigma; RRID:AB_477375), anti-Dsred (rabbit 1:1000, Clontech), anti-synaptotagmin 2 (1:1000, mouse, Znp-1 Zebrafish International Resource center; RRID:AB_10013783), anti-calbindin antibody (rabbit, 1:1000, Swant Inc; RRID:AB_10000340), anti-GABA$_A$β2/3 (mouse, 1:500 Millipore-Sigma) anti-GABA$_A$α1 receptor subunit (polyclonal guinea-pig, 1:5000, kindly provided by J.M.

Fritschy), and anti-GABA$_C$ρ receptor subunit (1:500, rabbit, kindly provided by R. Enz, H. Wassle, and S. Haverkamp). Retinas were thereafter incubated in secondary antibody solution using anti-iso-typic Alexa Fluor (1:1000, Invitrogen) conjugates. Retinas were finally mounted on slides with Vecta-shield antifade mounting medium (Vector Labs; RRID:AB_2336789).

## Electrophysiology

Electrophysiology experiments were performed on whole-mounted retinal preparations made from dark-adapted KO and WT mice. Mice were sacrificed via cervical dislocation and enucleation was subsequently performed. Retinas were isolated in oxygenated (95% O$_2$/5% CO$_2$) Ames medium (Sigma-Aldrich) at 32–34°C, mounted flat in a recording chamber and perfused with oxygenated Ames medium at a flow rate of ~8 mL/min during recordings. Retinas were mounted ganglion cell side up (*Sinha et al., 2016*; *Sinha et al., 2020*) for recordings. The retinas were embedded in aga-rose and sliced as previously described (*Hoon et al., 2015*; *Sinha et al., 2020*) for RBC recordings. Retinal neurons were visualized for patch-clamp recordings using infrared light (>900 nm). All record-ings were obtained from the ventral retina. Voltage-clamp recordings from RBCs and ONα GCs were made with pipettes (~10 MΩ for RBCs and 3–4 MΩ for ONα GCs) filled with an intracellular solution containing (in mM) 105 Cs methanesulfonate, 10 tetraethylammonium chloride, 20 HEPES, 10 EGTA, 2 QX-314, 5 Mg-ATP, 0.5 Tris-GTP (~280 mOsm, pH ~7.2 with KOH). For all voltage-clamp recordings, cells were held at estimated inhibitory and excitatory reversal potentials ~−60 mV and ~0 mV respectively in order to measure excitatory or inhibitory synaptic inputs. Absolute vol-tages were corrected for liquid junction potentials. For puff recordings of RBCs, GABA was applied with a Picospritzer II (General Valve) connected to a patch pipette with a resistance of ~5–7 MΩ. GABA (200 µM) was prepared in HEPES-buffered Ames medium with 0.1 mM Alexa 488 hydrazide. Puffing duration (50 ms) and direction were chosen such that the axon terminal of the RBC was completely covered by the puff. For the quantification of GABA-evoked currents, peak amplitude relative to the baseline current before stimulus/drug application was determined and averaged across cells. (1,2,5,6- Tetrahydropyridin-4-yl) methylphosphinic acid (TPMPA, 50 µM; Tocris, Bristol, United Kingdom) and GABAzine (20 µM; Tocris, Bristol, United Kingdom) were added to the perfu-sion solution for RBC recordings as indicated. Alexa 594 dye (100–200 µM) was added to the intra-cellular solution for the following image acquisition of the exemplar ONα GC shown in *Figure 2A* using the software ScanImage (RRID:SCR_014307) and analysed using the software ImageJ (RRID:SCR_003070). Light responses were recorded from ONα GCs using whole-cell and cell-attached recordings. LED light sources with peak spectral output at 360 or 405 nm respectively were used to deliver full-field light stimuli that were 500 µm in diameter and focused on the photoreceptor layer through the optics of the microscope. Photon densities were calibrated using estimations of opsin photoisomerisations per photoreceptor, assuming a rod collecting area of 0.5 µm$^2$ (*Field and Rieke, 2002*) and a cone collecting area of 0.2 µm$^2$ (*Nikonov et al., 2006*). Recordings were made in dark-ness or background light levels at which rods dominate retinal responses and light levels at which cones dominate (~1000 R*/S cone/s).

## Electrophysiology data acquisition and analysis

All electrophysiology data was low pass-filtered at 3 kHz, digitized at 10 kHz, and acquired using a Multiclamp 700B amplifier. The data was acquired using Symphony Data Acquisition Software, an open-source, MATLAB-based electrophysiology software (https://github.com/symphony-das). Sub-sequent data analysis was performed using self-written code in MATLAB (Mathworks; RRID:SCR_001622) and Igor Pro (WaveMetrics; RRID:SCR_000325). Peak response amplitudes of ONα GC were quantified by taking the peak spike rate or current during the stimulus presentation. The sustained component of ONα GC response to the 0.5 s light step was estimated by taking the average spike rate or excitatory current over a 200 ms time window (from 0.3 to 0.5 s) from the time of stimulus. The total spike response in *Figure 2C* was estimated as the average spike rate over the duration of the 0.5 s light step. Time to peak was estimated as the time taken for the response to reach from the stimulus onset to the peak amplitude. Decay time was estimated as the time taken for the response to recover from the peak amplitude to a value equal to the pre-stimulus baseline. The time of onset for the sustained phase in *Figure 3—figure supplement 1C–E* was estimated using two approaches – (i) on a per trace basis as the time after the stimulus onset to when the response

reached a fit line with zero slope and (ii) by fitting an exponential function to the response (average traces across trials from each cell) from the end of the transient phase (~175 ms from the time of stimulus onset) to the end of the stimulus and defining the onset time as the time at which the exponential fit decays to 20% of its initial value. Both analysis approaches yielded similar results (*Figure 3—figure supplement 1D,E*). Rebound amplitude for flash responses in *Figure 3M* was estimated as the amplitude of the peak of the overshoot from the baseline.

LN models (*Figure 4*) were derived from ONα GC responses to randomly varying light stimuli (Gaussian distribution of light intensities with standard deviation = 50% of mean intensity; 0–60 Hz bandwidth) as previously described (*Kim and Rieke, 2001*; *Rieke, 2001*; *Sinha et al., 2017*; *Sinha et al., 2016*). The linear temporal filter and the static nonlinearity were estimated using previously described methods (*Kim and Rieke, 2001*). Contrast-dependent changes are shared by the y-axis of the linear filter and the x-axis of the static nonlinearity. Therefore, to estimate the contrast gain unambiguously we needed to scale the linear filter in amplitude such that the variance of the filtered stimulus was equal to the variance of the stimulus. By scaling the filter in this way, differences in the contrast gain of the model would be reflected in changes in the slope of the static nonlinearity. The slope of the static nonlinearity was estimated as the average slope within the linear region of a quintic polynomial fit to each response in KO and WT retina. The slopes estimated from fits with a cumulative density function yielded similar values (data not shown).

## Confocal microscopy and image analysis

A Leica SP8 confocal microscope and a 1.4 NA 63× oil immersion objective at a voxel size of around 0.05–0.05–0.3 μm (x–y–z) was used to acquire the images in this study. The image stacks acquired were further processed in Amira (Thermo Fisher Scientific; RRID:SCR_007353) software. BC processes were masked in 3D using the *LabelField* function of Amira. Following masking, the receptor signal within the BC process was isolated using the Amira *Arithmetic* tool. A threshold was subsequently applied in order to eliminate background receptor signals and the volume of receptor pixels was expressed as a percent occupancy relative to the volume of the BC processes (for details see *Hoon et al., 2015*; *Hoon et al., 2017*). A similar procedure was carried out to isolate the GABA$_A$β2/3 receptor signal within calbindin positive AC and GC processes and ascertain percentage volume occupancy of this receptor signal.

## Statistical analysis

We used the unpaired two-tailed t-test for all the statistical analysis. Error bars indicate SEM. The significance threshold was placed at α = 0.05 (n.s., p>0.05; *p<0.05; **p<0.01; ***p<0.001). In all figures, 'n' refers to the number of cells analyzed except in *Figure 1C and D* and *Figure 1—figure supplement 1E* where 'n' refers to number of retinas analyzed.

## Acknowledgements

We thank JM Fritschy, R Enz, H Wässle, and S Haverkamp for their generous gifts of GABA receptor antibodies. We thank Rachel Wong and B Luscher for the *Gabrg2* floxed mutant mice. We thank Mike Ahlquist and Austin Butala for their technical support and all members of the lab for their helpful feedback on this project.

## Additional information

### Funding

| Funder | Grant reference number | Author |
|---|---|---|
| National Eye Institute | EY026070 | Raunak Sinha |
| University of Wisconsin-Madison | David and Nancy Walsh Family Professorship in Vision Research | Raunak Sinha |
| University of Wisconsin-Madison | Rebecca Brown Professorship in Vision Research | Mrinalini Hoon |

| Research to Prevent Blindness | | Mrinalini Hoon |
| --- | --- | --- |
| National Institute of General Medical Sciences | T32 Graduate Student Fellowship | Jenna Nagy |
| National Eye Institute | EY031677 | Mrinalini Hoon |
| National Institute of Neurological Disorders and Stroke | T32NS105602 | Briana Ebbinghaus |

The funders had no role in study design, data collection and interpretation, or the decision to submit the work for publication.

### Author contributions
Jenna Nagy, Formal analysis, Writing - original draft, Writing - review and editing, Investigation; Briana Ebbinghaus, Investigation; Mrinalini Hoon, Data curation, Formal analysis, Investigation, Methodology, Writing - original draft, Project administration, Writing - review and editing; Raunak Sinha, Conceptualization, Data curation, Formal analysis, Supervision, Funding acquisition, Investigation, Methodology, Writing - original draft, Writing - review and editing

### Author ORCIDs
Raunak Sinha ⓘ https://orcid.org/0000-0002-7553-1274

### Ethics
Animal experimentation: All experiments and animal care were conducted in accordance with the Institutional Animal Care and Use Committee (IACUC) of the University of Wisconsin-Madison and the National Institutes of Health. The protocol was approved by the Animal Care and Use Committee of the University of Wisconsin-Madison (Protocol ID: M006031-R01).

### Decision letter and Author response
Decision letter https://doi.org/10.7554/eLife.60994.sa1
Author response https://doi.org/10.7554/eLife.60994.sa2

## Additional files

### Supplementary files
• Transparent reporting form

### Data availability
All source data shown in the figures including some raw data are available in the data repository, Dryad, accessible via this link https://doi.org/10.5061/dryad.bg79cnpb4. Raw datasets are quite large in size and will be made available upon request by the corresponding author.

The following dataset was generated:

| Author(s) | Year | Dataset title | Dataset URL | Database and Identifier |
| --- | --- | --- | --- | --- |
| Nagy J, Ebbinghaus B, Hoon M, Sinha R | 2021 | GABAA presynaptic inhibition regulates the gain and kinetics of retinal output neurons | https://doi.org/10.5061/dryad.bg79cnpb4 | Dryad Digital Repository, 10.5061/dryad.bg79cnpb4 |

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
