## [Decision Letter]

**Acceptance summary:**

All three reviewers were impressed by the additional experiments, analysis and revisions to the text. As a result the manuscript is much improved. This is now a more definitive study on how GABA-A receptor mediated presynaptic inhibition impacts many features of light encoding.

**Decision letter after peer review:**

Thank you for submitting your article "GABA_A_ presynaptic inhibition regulates the gain and kinetics of retinal output neurons" for consideration by *eLife*. Your article has been reviewed by 3 peer reviewers, one of whom is a member of our Board of Reviewing Editors, and the evaluation has been overseen by Richard Aldrich as the Senior Editor. The following individual involved in review of your submission has agreed to reveal their identity: Katrin Francke.

The reviewers have discussed the reviews with one another and the Reviewing Editor has drafted this decision to help you prepare a revised submission.

Below is a synthesis of comments from all three reviewers. Note some of the comments are addressing the same point (namely interpretation of Figures 3 and 4). I thought it would be of benefit that you read the reviewers comments as they were written.

Summary:

The authors use targeted knockout approach to investigate the role of GABA_A_ receptor mediated presynaptic inhibition at the bipolar cell (BC) axon terminal on the output of a specific retinal ganglion cell (RGC) type. Using a Grm6-Cre mouse line crossed with a floxxed gamma2 subunit of the GABA-A receptor, they generated mice which eliminated GABA-A receptors from ON-cone bipolar cells and rod bipolar cells. This strategy was used previously (Hoon et al., PNAS, 2015) to eliminate of GABA-A receptors on cone bipolar cell terminals while maintaining GABA-A receptors in the bipolar cell dendrites. They confirm the KO of GABA-A receptors at terminal by immunohistochemistry and functional recordings. Then, they focus on how a lack of GABA_A_ receptor mediated presynaptic inhibition modulates the output of On α RGCs – a well-described circuit of the inner retina. They find that this selective loss of GABA_A_ receptors enhances sensitivity for both dim and bright light conditions and that it affects the kinetic and contrast response of light-evoked, cone-driven responses of On α RGCs – thereby demonstrating that presynaptic inhibition shapes the output of a distinct retinal circuit.

Essential revisions:

All three reviewers felt the experiments were well executed and the results were presented clearly. However all noted that the implications of the data should be analyzed and discussed more thoroughly. Moreover, a few addition experiments were requested so that authors can be more confident that there was no impact on GABA-A receptors in amacrine cells or RGCs. All felt these issues must be addressed in order to increase the impact of the study.

1. The authors need to test more rigorously whether the KO has impacted GABA-A receptors on RGCs and ACs. The authors show a dramatic decrease in GABA-A receptors in targeted KOs in the ON sublamina (Figure 1). However, there should still be GABA-A receptors on ACs and RGCs. Perhaps they are at a lower density and difficult to see but the other possibility is that there has been a development downregulation of postsynaptic GABA-A receptors in these KOs. Hoon et al. has shown that reduction of GABA-A transmission can impact maturation of GABA-A receptor in bipolar cell terminals so this might also happen in RGC or AC dendrites. These are important control to make sure that their results are indeed due to loss of presynaptic inhibition at the BC synapse and not due to loss of serial inhibition at amacrine cell processes or loss of inhibition on RGCs.

2. The authors make the claim that the increases in firing rate observed in KO mice is due to increases in excitatory currents and therefore due entirely to loss of presynaptic inhibition on bipolar cell terminals. To confirm this conclusion, they should assay the impact of inhibition in KO. They present what feels like a thin data set addressing this issue in Figure 2 – supplement 1 – with one example cell and no summary data. This data set needs to be expanded to make this conclusion.

3. The authors demonstrate that eliminating GABA_A_ mediated presynaptic inhibition at the On BC synapse increases the gain and sensitivity of excitatory inputs to On α RGCs (Figures 2, 3) – which is likely not surprising given previous studies, but corresponds to a reasonable proof-of-concept result. However, when they continue to further investigate the functional relevance of the reduced presynaptic inhibition on On α RGC signaling, the interpretation of the results is incomplete (see next two points 3.1, 3.2). Here, a more detailed interpretation and added context is critical for the impact of the study – especially since the results about increased sensitivity are rather expected and in general the authors use a relatively simple stimulus set.

3.1. In Figure 3F, the authors show the mean excitatory synaptic current to a brief flash for WT and KO mice and quantify the change in response amplitude (Figure 3G). However, not only the amplitude but also the response shape looks significantly different, with the KO trace showing a rebound that is not present in the WT trace. This is an interesting finding that the authors do not comment on. The rebound could be due to a decrease in spontaneous activity of BCs, which would be visible in single trial responses. Is this what the authors observed? This effect is also present in traces shown in Figure 3C and might be related to the delayed onset of the sustained response component in Figure 3B and Figure 3C. I suggest the authors quantify the rebound and the onset of the delayed response component and comment and discuss this finding – what might be the underlying mechanism and the functional relevance?

3.2. In Figure 4, the authors find that in KO animals, temporal kernels estimated from a full-field, white noise stimulus exhibit a slower kinetic and that contrast changes are encoded by a larger current range. However, I do not follow their conclusion: "This means that the ONα GCs in the KO retina are sensitive to a larger (~two-fold) range of contrasts than the ONα GC in the WT retina". Instead, I think that their results show that the same range of contrasts is encoded by a larger current range, suggesting that in KO animals ONα GCs have a larger dynamic range which likely allows better discrimination of contrast levels. Also, the authors do not speculate about the functional implications of their finding. Their results suggest that GABA_A_ receptor mediated inhibition at On BC axon terminals speeds up ONα GC responses, however, at the cost of discriminating different contrast levels. Therefore, one could speculate that this circuit prioritizes fast changes over contrast discrimination – would the authors agree with this interpretation? For which visual tasks might this be useful?

4. Figures 2-3 The authors have interesting data here that should be investigated more thoroughly. It makes sense that removing GABA_A_ receptor mediated inhibition from ON bipolar cells would increase the amplitude of responses in the ONαGCs, as is shown here. However, as the authors discuss later, removing inhibition also likely affects the timing of responses. The authors have data on the spike and currents responses to a brief flash of light that should give interesting information about kinetics, but this is not analyzed. Previous data (Eggers and Lukasiewicz, J Neuroscience 2006) suggests that removing GABA_A_ specifically in ON bipolar cells would affect the timing of the peak more than the decay/half-width. Does that happen here?

5. Figure 4. The temporal differences in the linear filter are interesting. It appears that both the time to peak and the half-width of the filter are changed in the examples shown. Was this true generally? Also, did the change in the temporal filter correlate with a change in the time to peak and decay time/half width of the responses to brief light stimuli? This should be similar to what is being estimated by the linear filters. This could be an interesting point in the paper because often the brief flash response and linear filters are not measured in the same paper, in the same types of cells. Also, please show the similar data in Figure 4A, B, D for the rod activated responses. Rod vs. cone activation is an interesting comparison, especially as GABA_A_ receptors may have a larger role in ON cone bipolar cells than in rod bipolar cells.

6. The authors need to clarify their stats. They use SEM to represent the variance in their data , implying normal distributions. However, they use rank sum tests for significance, implying non-normal data. The authors need to clarify whether their data follows a normal distribution and if so, then they should use more powerful t-tests rather than rank tests. If their data does not follow a normal distribution, then they need to use SD to represent variance or confidence intervals to represent significance.

---

## [Author Response]

Essential revisions:All three reviewers felt the experiments were well executed and the results were presented clearly. However all noted that the implications of the data should be analyzed and discussed more thoroughly. Moreover, a few addition experiments were requested so that authors can be more confident that there was no impact on GABA-A receptors in amacrine cells or RGCs. All felt these issues must be addressed in order to increase the impact of the study.1. The authors need to test more rigorously whether the KO has impacted GABA-A receptors on RGCs and ACs. The authors show a dramatic decrease in GABA-A receptors in targeted KOs in the ON sublamina (Figure 1). However, there should still be GABA-A receptors on ACs and RGCs. Perhaps they are at a lower density and difficult to see but the other possibility is that there has been a development downregulation of postsynaptic GABA-A receptors in these KOs. Hoon et al. has shown that reduction of GABA-A transmission can impact maturation of GABA-A receptor in bipolar cell terminals so this might also happen in RGC or AC dendrites. These are important control to make sure that their results are indeed due to loss of presynaptic inhibition at the BC synapse and not due to loss of serial inhibition at amacrine cell processes or loss of inhibition on RGCs.

The reviewer has raised an excellent point about the specificity of the GABA_A_ receptor deletion in the conditional knockout (KO) mice. We have addressed this concern by performing additional control experiments. We have assessed if expression of GABA_A_ receptors in ACs and RGCs processes in the KO retina is changed compared to that in the WT retina through new immunohistochemistry experiments that are now included in Figure 1—figure supplement 1D,E. By performing co-labeling for calbindin, a marker for AC and GC processes in the inner plexiform layer (IPL), and for specific GABA_A_ receptor subunits, we found that the expression of GABA_A_ receptors in the AC and GC processes in the ON sublamina of the IPL remains unchanged in the KO retina (Figure 1—figure supplement 1D,E). We have in addition tested if postsynaptic inhibition on ONa GCs is altered in the KO retina (Figure 3—figure supplement 2). We find no difference in the amplitude of the light-evoked inhibitory current recorded from ONa GCs in KO and WT retina.

2. The authors make the claim that the increases in firing rate observed in KO mice is due to increases in excitatory currents and therefore due entirely to loss of presynaptic inhibition on bipolar cell terminals. To confirm this conclusion, they should assay the impact of inhibition in KO. They present what feels like a thin data set addressing this issue in Figure 2 – supplement 1 – with one example cell and no summary data. This data set needs to be expanded to make this conclusion.

We have increased the sample size and added a bar plot with quantification of the peak amplitude of the light-evoked inhibitory current for ONa GCs in KO and WT retina (Figure 3—figure supplement 2). This revised dataset shows that post-synaptic inhibition received by the ONa GCs remains unchanged in the KO retina.

3. The authors demonstrate that eliminating GABA_A_ mediated presynaptic inhibition at the On BC synapse increases the gain and sensitivity of excitatory inputs to On α RGCs (Figures 2, 3) – which is likely not surprising given previous studies, but corresponds to a reasonable proof-of-concept result. However, when they continue to further investigate the functional relevance of the reduced presynaptic inhibition on On α RGC signaling, the interpretation of the results is incomplete (see next two points -- 3.1, 3.2). Here, a more detailed interpretation and added context is critical for the impact of the study – especially since the results about increased sensitivity are rather expected and in general the authors use a relatively simple stimulus set.

We thank the reviewer for raising this concern about interpretation of results and added context. We have now included several new analyses, as the reviewers suggested (see below), which have bolstered the results and the impact of the study. We have also expanded the Discussion section substantially to provide better context for how our current findings advance the understanding of presynaptic inhibition and its role in retinal function.

3.1. In Figure 3F, the authors show the mean excitatory synaptic current to a brief flash for WT and KO mice and quantify the change in response amplitude (Figure 3G). However, not only the amplitude but also the response shape looks significantly different, with the KO trace showing a rebound that is not present in the WT trace. This is an interesting finding that the authors do not comment on. The rebound could be due to a decrease in spontaneous activity of BCs, which would be visible in single trial responses. Is this what the authors observed? This effect is also present in traces shown in Figure 3C and might be related to the delayed onset of the sustained response component in Figure 3B and Figure 3C. I suggest the authors quantify the rebound and the onset of the delayed response component and comment and discuss this finding – what might be the underlying mechanism and the functional relevance?

The reviewer raises an interesting point about the response rebound and its underlying implications. As per reviewers’ suggestion, we quantified both the rebound amplitude for the response to brief 10 ms light flashes as well as the time of onset of the sustained phase for the responses to a light step (0.5s duration) in Figure 3. We have added the new analysis in Figure 3 (Figure 3M and Figure 3—figure supplement 1D). We do not see a significant difference for either of those parameters across **ONa GCs** between KO and WT retina (Figure 3M and Figure 3—figure supplement 1D,E). To better represent the data, we have replaced exemplar traces in all figures with average responses (across cells). A previous study has shown that lack of GABA_C_ receptor mediated inhibition enhances glutamate release from bipolar cells which leads to spillover activation of perisynaptic NMDA receptors on ON GCs and thus extends their dynamic range (Sagdullaev et al., 2006). We agree with the reviewer that it will be interesting to determine if a similar synaptic mechanism is at play in absence of presynaptic GABA_A_ receptors for enhancing output to ONa GCs. This will be a subject of future investigation using detailed pharmacology and recordings of AMPA and NMDA receptor mediated spontaneous/evoked excitatory postsynaptic currents. We have added this potential mechanism of regulating synaptic output in the revised discussion (lines 367-373).

3.2. In Figure 4, the authors find that in KO animals, temporal kernels estimated from a full-field, white noise stimulus exhibit a slower kinetic and that contrast changes are encoded by a larger current range. However, I do not follow their conclusion: "This means that the ONα GCs in the KO retina are sensitive to a larger (~two-fold) range of contrasts than the ONα GC in the WT retina". Instead, I think that their results show that the same range of contrasts is encoded by a larger current range, suggesting that in KO animals ONα GCs have a larger dynamic range which likely allows better discrimination of contrast levels. Also, the authors do not speculate about the functional implications of their finding. Their results suggest that GABA_A_ receptor mediated inhibition at On BC axon terminals speeds up ONα GC responses, however, at the cost of discriminating different contrast levels. Therefore, one could speculate that this circuit prioritizes fast changes over contrast discrimination – would the authors agree with this interpretation? For which visual tasks might this be useful?

We thank the reviewer for pointing this out. We have corrected this and discussed the interpretation and functional relevance at length in the expanded/revised Discussion section. We have dedicated one part of the discussion on the role of presynaptic inhibition in contrast encoding and temporal filtering for the ONa GC circuit. To better estimate the contrast sensitivity from the white noise stimuli, we have re-analyzed the nonlinearity after scaling the linear filter (see methods) since contrast-dependent changes are shared by both the y-axis of the linear filter and the x-axis of the nonlinearity. After scaling the filters, the slope of the nonlinearity can be used as a measure of contrast gain (Figure 4E, G, K, M). This clearly shows that ONα GC contrast sensitivity is increased in the KO retina under both rod and cone activated light conditions (Figure 4G, M). Our results thus suggest that GABA_A_ presynaptic inhibition limits the size of the ONα GC current response to contrasts to perhaps allow for encoding over a broader range of contrast without saturation. We are a bit hesitant to speculate that the ONα GC circuit prioritizes fast changes over contrast discrimination. What if, perhaps, the variance in light encountered at this given light background in the real world was such that the animal needed to discriminate over a wide range of contrasts. Then, having the lower contrast sensitivity is actually a benefit, since it is matched to the statistics of the inputs.

4. Figures 2-3 The authors have interesting data here that should be investigated more thoroughly. It makes sense that removing GABA_A_ receptor mediated inhibition from ON bipolar cells would increase the amplitude of responses in the ONαGCs, as is shown here. However, as the authors discuss later, removing inhibition also likely affects the timing of responses. The authors have data on the spike and currents responses to a brief flash of light that should give interesting information about kinetics, but this is not analyzed. Previous data (Eggers and Lukasiewicz, J Neuroscience 2006) suggests that removing GABA_A_ specifically in ON bipolar cells would affect the timing of the peak more than the decay/half-width. Does that happen here?

We thank the reviewer for making this excellent suggestion. We have now analyzed the kinetics of the light-evoked spike and excitatory current to a brief light flash and a light step (0.5s duration). We have further quantified the decay times of the excitatory current responses to light flash in Figure 3. We have added bar plots for each of the above kinetic analysis in Figures 2E,I and 3D,H,K and L. The results we observed for the linear filters are consistent with that for the responses to light flashes/steps. The differences in the kinetics of excitatory inputs of ONα GCs for cone-driven signals between KO and WT retina and the similarities in the kinetics of excitatory inputs of ONα GCs for rod-driven signals between KO and WT retina is reflected in the ONα GC spike output (Figure 2E and 3D). We see similar increase (~10 ms) in time to peak and decay time in the light-evoked excitatory inputs of ONα GCs in the KO retina compared to that in the WT retina (Figure 4C vs Figure 4D; Figure 3K vs 3L).

5. Figure 4. The temporal differences in the linear filter are interesting. It appears that both the time to peak and the half-width of the filter are changed in the examples shown. Was this true generally? Also, did the change in the temporal filter correlate with a change in the time to peak and decay time/half width of the responses to brief light stimuli? This should be similar to what is being estimated by the linear filters. This could be an interesting point in the paper because often the brief flash response and linear filters are not measured in the same paper, in the same types of cells. Also, please show the similar data in Figure 4A, B, D for the rod activated responses. Rod vs. cone activation is an interesting comparison, especially as GABA_A_ receptors may have a larger role in ON cone bipolar cells than in rod bipolar cells.

This is an excellent point raised by the reviewer. See response to reviewer comment 4. The results we observed for the linear filters are consistent with that for the responses to brief light flashes. We have revised Figure 4 and added similar data for rod-mediated responses.We have included in the Discussion section a paragraph about this difference in temporal filtering between rod and cone-mediated signals and why GABA_A_ presynaptic inhibition does not seem to play a role in shaping the time-course of rod-mediated signals in the ONα GC circuit (lines 322-352).

6. The authors need to clarify their stats. They use SEM to represent the variance in their data , implying normal distributions. However, they use rank sum tests for significance, implying non-normal data. The authors need to clarify whether their data follows a normal distribution and if so, then they should use more powerful t-tests rather than rank tests. If their data does not follow a normal distribution, then they need to use SD to represent variance or confidence intervals to represent significance.

We thank the reviewer for pointing this out. We have redone all the statistical analysis using an unpaired two-tailed t-test. We have revised the text in the methods accordingly. We have shown all data points for sample size of n<10.